

# Combined Hazard Analysis of Flood and Tsunamis on The Western Mediterranean Coast of Turkey

Cuneyt Yavuz [1,2], Kutay Yilmaz [3], Gorkem Onder [4]

[1]Department of Construction Technologies, Technical Sciences Vocational School, Dumlupinar University, 43000, Kutahya,
Turkey
[2]Graduate School of Engineering, The University of Tokyo, 113-8654, Tokyo, Japan
[3]ALTER International Engineering Inc. Co., 06800, Ankara, Turkey
[4] Sumodel Engineering Inc. Co., 06800, Ankara. Turkey

*Correspondence to*: Cuneyt Yavuz (cuneyt.yavuz@dpu.edu.tr)

**Abstract.** Flood has always been a devastating hazard for social and economic assets and activities. Especially, low-land areas such as coastal regions can be more vulnerable to inundations. The combination of different natural hazards observed at the same time is definitely worsening the situation in the affected regions. The goal of this study is to conduct a distinctive combined hazards analysis considering flood hazards with the contribution of potential earthquake-triggered tsunamis that
might be observed through Fethiye coastline and city center. For this purpose, tsunami hazard curves are generated based on Monte Carlo Simulations. Comprehensive stochastic hazard analyses are performed considering aleatory variability of earthquake-triggered tsunamis and epistemic uncertainty of flood having 10 year return period. Numerical simulations are conducted to combine the potential tsunamis and flood events that are able to adversely affect the selected region. The results of this study show that the blockage of stream outlets due to tsunami waves drastically increases the inundated areas and
worsens the condition for the selected region.

**Keywords:** Stochastic analysis; Monte Carlo simulation; tsunami simulation; flood; potential combined hazard assessment

## 1 Introduction

Flood hazards have been one of the most destructive and frequent global-wide natural hazards resulting in loss of lives, livestock, and economic assets (Slater&Villarini, 2016; Alfieri et al., 2017; Kreibich et al., 2017; Qiang, 2019; Zhai et al.,
2020). Even though low-land and plain areas where 80% of the world population live can create an easy way for urbanization, they also vulnerable to flood risk and the hazardous effects of floods will increase in the future due to the changing hydrological cycle in recent years (Lamond et al., 2011). As the number of flood hazards increases, the amounts of flood losses are going to follow a parallel trend, accordingly. Hemmati et al. (2020), stated that both the number of floods and destructive economic results have been drastically increased since the 1990s (see Figure 1).


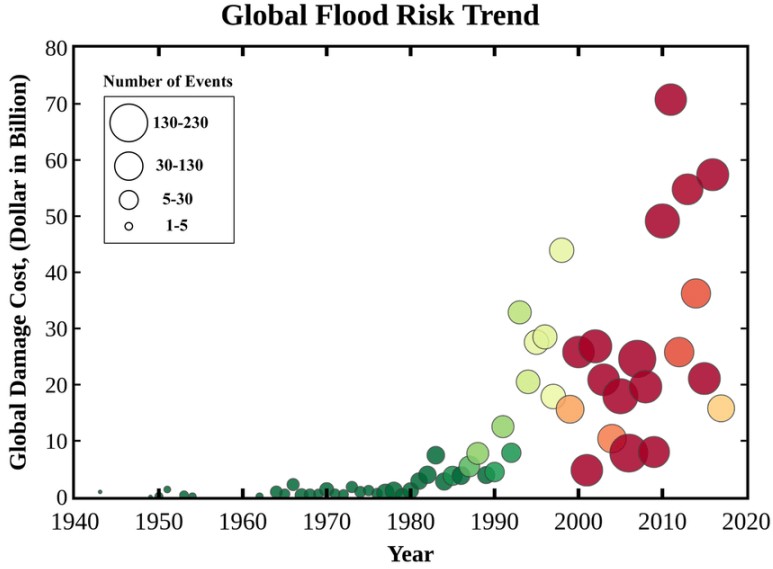

**Figure 1.** Number of floods and destructive economic results at global scale (Munich Re, 2020)

Independent from flood hazard, the tsunami which can be a long or short-term event is rare but can cause catastrophic damage to economic and social assets and activities (Wolfgang, 2005; Kundzewicz et al., 2017; Subyani et al., 2017; Fukao, 1979). Devastating economic losses and loss of lives have been recorded for the countries that experienced tsunami events, especially for the last two decades (Nadim&Glade, 2006; Carreño et al., 2007; Cardona et al., 2010; Sørensen et al., 2012; Lane et al., 2013; Horspool et al., 2014; Goda&Abilova, 2016). Scientists have revealed significant and reliable hazard evaluation methods for tsunami hazard assessment according to adverse consequences of the experienced tsunamis (Jelínek et al., 2012).

Combined hazard assessment of floods with different natural hazards can be found in the literature. For instance, climate change-related flood hazard assessment has been widely investigated (Blöschl et al., 2017; Kaspersen et al., 2017; Szewrański et al., 2018; Carter et al., 2018; Barkey et al., 2019). However, the investigations covering simultaneous assessment of flood and tsunami events have been limited. Even if the coincidence of flood and tsunami hazards may be experienced once in a blue moon, it should also be investigated due to the uncertainty of the time of occurrence for these natural hazards. The objective of this study is to reveal a statistical methodology to evaluate the aggregate potential hazard levels due to flood hazards with the presence of earthquake-triggered tsunamis.

As commonly used issues in stochastic hazard analysis of any kind of hazard in the literature (Bommer, 2003; Helton et al., 2010), aleatory and epistemic uncertainties are considered to generate combined hazard analysis in this study. The exceedance of flood hazard is strongly likely depending on geological and meteorological circumstances, the hazard is included in the stochastic analyses conducted in this study as aleatory variability. Since the occurrence of the tsunami is generally rare compared with flood hazards, tsunami events are inspected by considering epistemic uncertainty in this study. Additionally, hypothetical earthquake magnitudes $M_w$ are generated using Monte Carlo simulations to obtain a required number of random earthquake sources in the bathymetry.



The proposed methodology is applied to Fethiye city center which is one of the most popular touristic destinations on the Western Mediterranean coast of Turkey. The selection of this site is based on the documented 7 tsunami events throughout the history and evidences of tsunami deposits found by the researchers (Cita&Rimoldi, 1997; Papadopoulos, 2009; Altinok et al.,

2011) around Fethiye Bay. Fethiye coastline was hit several times with destructive tsunami waves reaching up to 1.8 m and significant inundation distances were recorded (Papadopoulos, 2009). Location of the study area for the case study is shown in Figure 2.

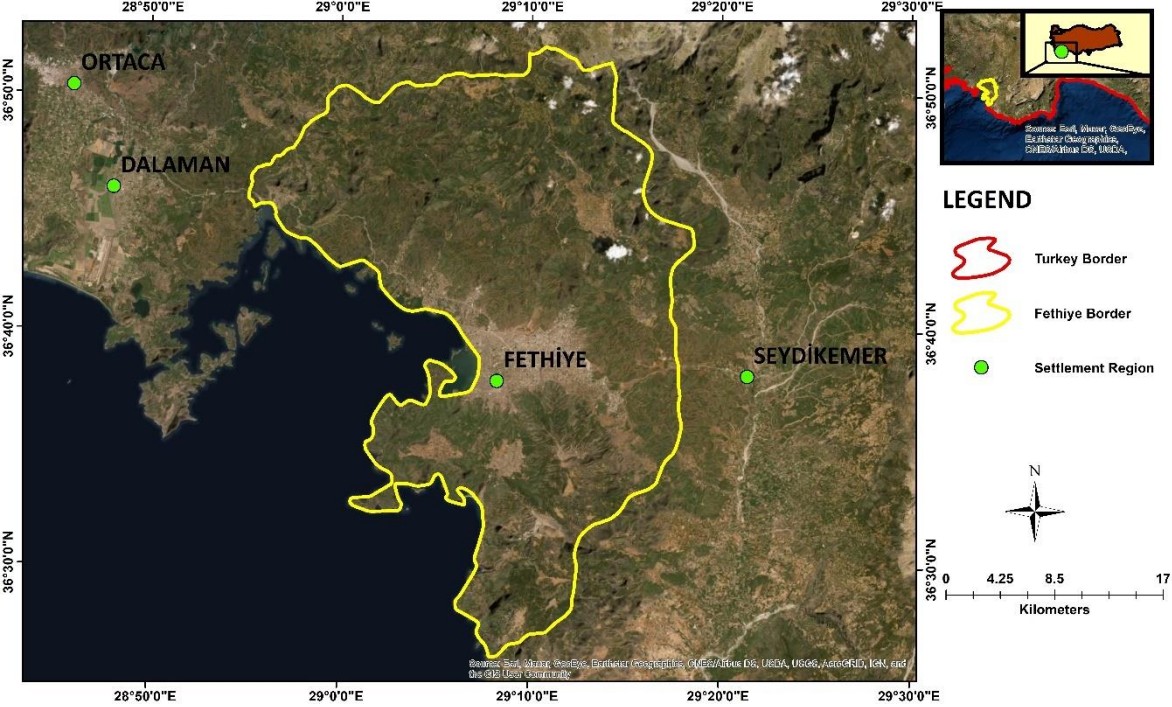

**Figure 2.** Study area and its location on satellite image (Source: Esri, Maxar, GeoEye, Earthstar Geographics, CNES/Airbus
DS, USDA, USGS, AeroGRID, IGN, and the GIS User Community).

## 2 Materials and Methods

Probabilistic combined hazard assessment approach (PCHA) is applied in this study. By doing so, the two dynamic natural hazards are aimed to evaluate one by one and simultaneously. 523 historical earthquakes recorded between 1900-2013 are retrieved from European Union funded Tsunami risk and strategies for the European region (TRANSFER, n.d.) project

catalogue. Gutenberg-Richter relationship is used to determine the best-fitted distribution for the historical earthquake magnitudes. Tsunami hazard curves are generated based on the hypothetical earthquake magnitudes ($M_w$) produced from 100000 Monte Carlo simulations. Nami-Dance software is used to simulate the hypothetical earthquakes having the $M_w \geq 6.5$ (USGS, n.d.) and resulting in tsunami wave heights are computed at the coast of Fethiye city center.




Flood hazard having the recurrence period of 10 years ($Q_{10}$), on the other hand, is modeled by MIKE 11, MIKE 21 FM and

MIKE Flood considering with and without tsunami wave existence at the coasts (DHI, 2016). As a more frequent flood period, $Q_{10}$ is evaluated in this study instead of the flood events having the return period of 50 years or 100 years. Thus, hazard levels considering both flood, earthquake-triggered tsunami, and tsunami-drifted flood hazards can be compared for the selected region. The inundation levels presented in this study have just resulted from the numerical analysis of both hazards. Potential hazard that can be resulted due to seismicity are not in the scope of this study. The flowchart of the methodology used in this

study is illustrated in Figure 3.

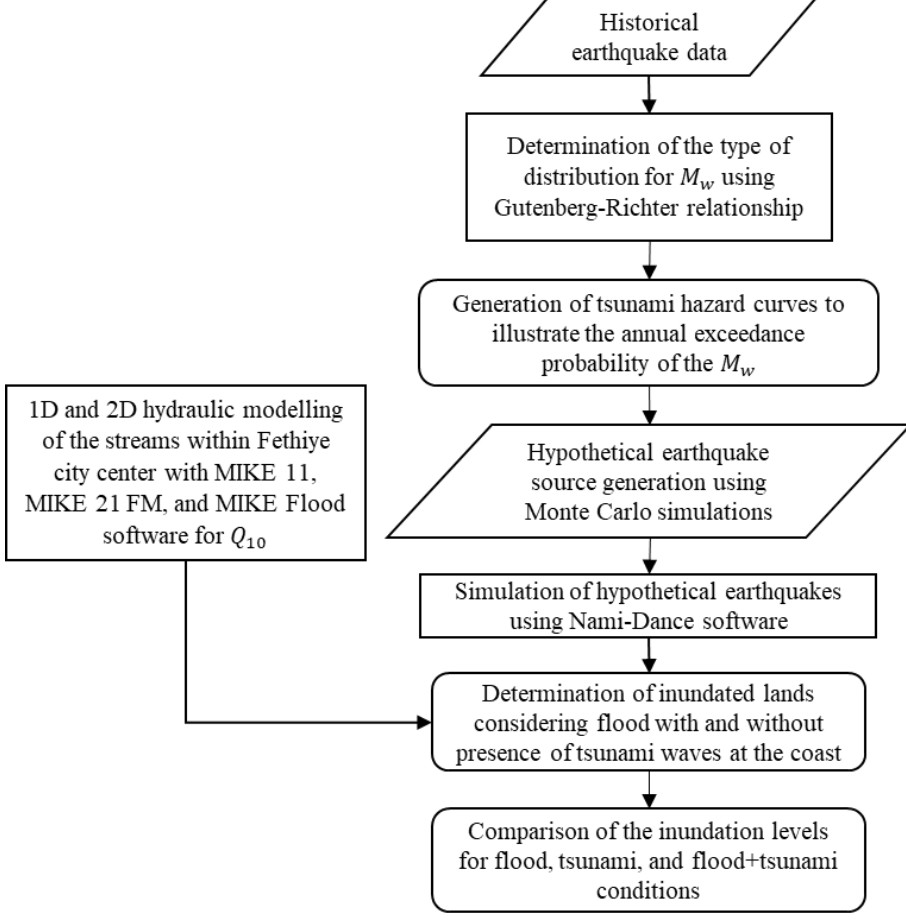

**Figure 3.** Combined hazard assessment framework used in this study

## 2.1. Generation of Hypothetical Earthquakes

Random $M_w$ are generated using Monte Carlo simulation, also known as stochastic modeling is accepted as one of the most

flexible and easiest methods to implement probabilistic hazard analysis (Ferson, 1996). Probability density function is defined for $M_w$ that defined as the independent parameter of the earthquake. Normal distribution is assigned to $M_w$ depending on the


probability density function. Kolmogorov-Smirnov test is applied to the assigned distribution to test the goodness of fit via p-value. Feasibility of $M_w$ data production is satisfied by conducting 100000 Monte Carlo simulations. Sufficiency of the generated data and the consistency of normal distribution are inspected using Gutenberg-Richter Relationship. For Fethiye

bay, the a and b values used in the Gutenberg-Richter relationship are obtained from Pamukcu et al., (2021) as 4.6624 and 0.8644, respectively. QQ plot obtained from Gutenberg-Richter relationship for the study area is illustrated in Figure 4. For the moment magnitudes greater than 6.0 illustrated in Figure 4, the normal distribution has a good coincidence with the Gutenberg-Richter relation.

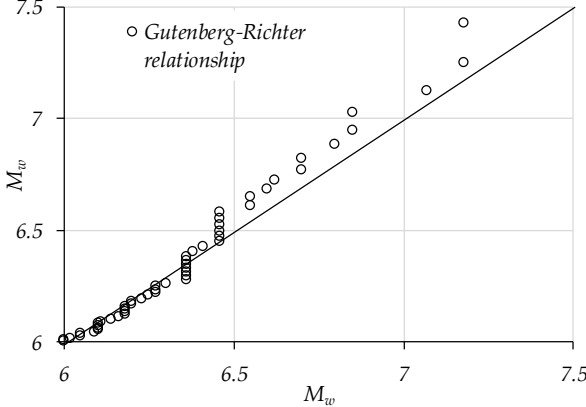

90                **Figure 4.** QQ plot of $M_w$ for Gutenberg-Richter law and the normal distribution.

Tsunami hazards curves are derived to determine the reliability of Monte Carlo simulations by considering the epistemic uncertainty of each hypothetical earthquake magnitude by checking the consistency of the curves (see Figure 5).

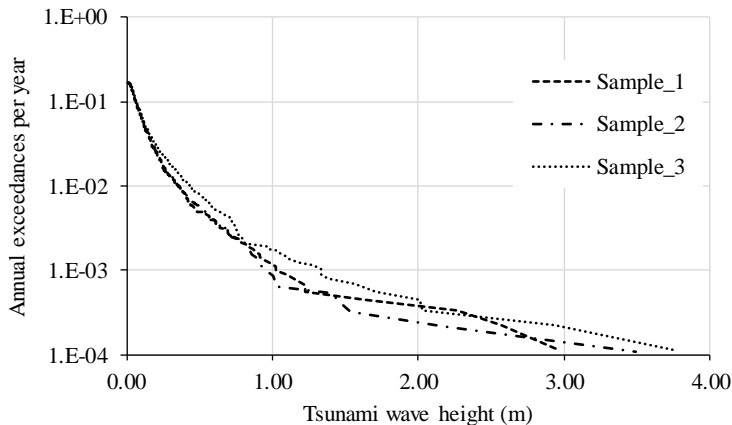

**Figure 5.** Tsunami hazard curves derived from 100000 Monte Carlo simulations.

Coincidence between the randomly generated $M_w$ shows that 100000 Monte Carlo simulations are sufficient up to $10^{-4}$/year annual exceedance of the tsunamigenic earthquake. As clearly stated in the literature, earthquakes having $M_w \geq 6.5$ can be considered tsunamigenic earthquakes (USGS, n.d.). Depending on this statement, 1561 out of 100000 randomly generated $M_w$




has a magnitude greater than 6.5 and is considered tsunamigenic earthquakes in this study. The generation steps of the hypothetical earthquake sources are given in Figure 6.

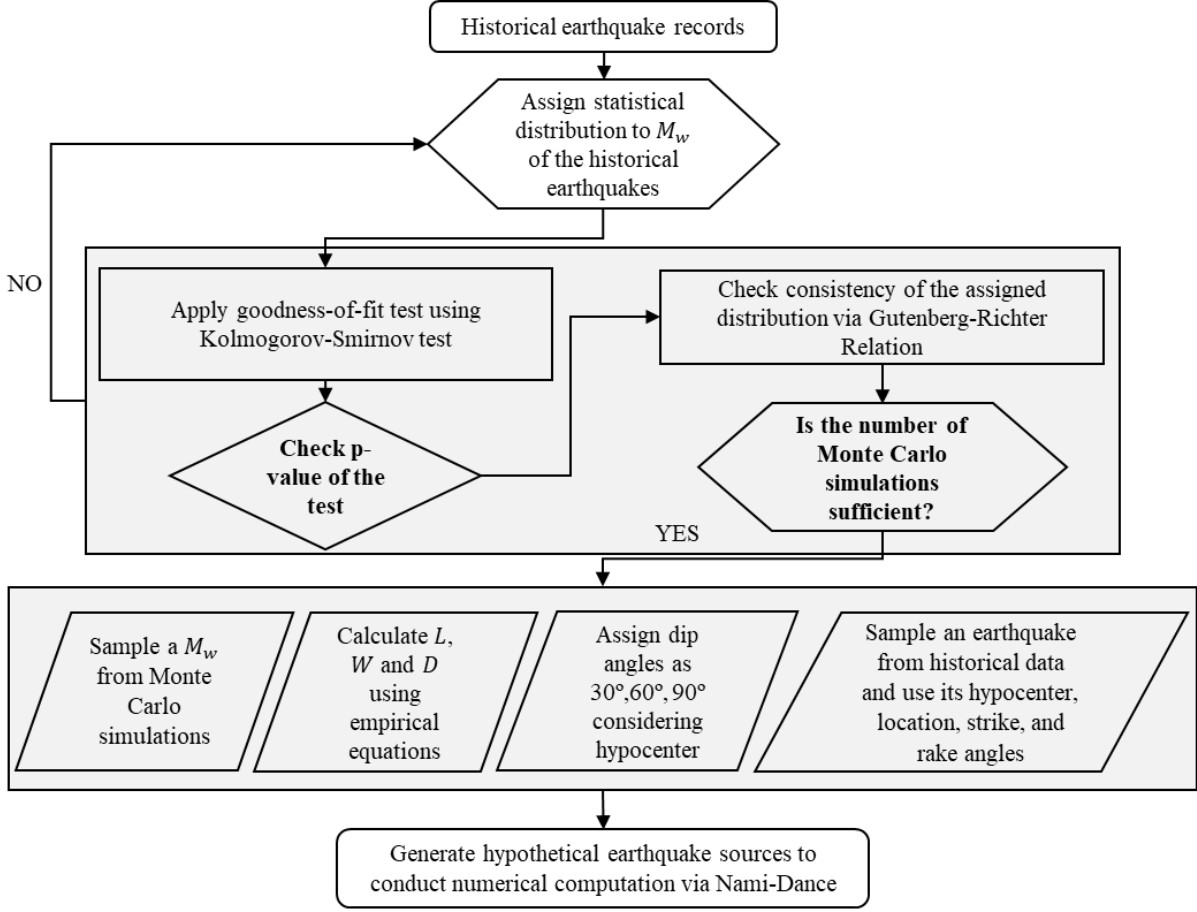


**Figure 6.** Tsunami hazard curves derived from 100000 Monte Carlo simulations.

The calculation procedure of the parameters of the hypothetical earthquake is explained, respectively. Fault length ($L$) of the hypothetical earthquake is calculated using the following equation (武村雅之, 1998):

$$logL = 0.5M_w - 1.91 \ for \ M_w < 6.8 \tag{1}$$

$$logL = 0.75M_w - 3.77 \ for \ M_w \geq 6.8 \tag{2}$$

The fault width ($W$) can then be calculated using the simple equation given for the rupture area ($S$) as $W = S/L$. Displacement

($D$) is also calculated using the empirical equation provided by Hanks&Kanamori, (1979):

$$M_w = {}^2\!/_3 \, log(M_0) - 10.7 \tag{3}$$

$$M_0 = \mu LWD \tag{4}$$

where $\mu$ is the shear modulus of crust ($3.43 * 10^{10} \ N/m^2$).
Depending on the hypocenter of the hypothetical earthquake, dip angles are assigned as $30^0, 60^0$, and $90^0$. The rest of the parameters are obtained directly from a sampled historical earthquake from the catalogue. The locations of the historical earthquakes are randomly assigned as the epicenter of the hypothetical earthquakes and are illustrated in Figure 7. Then, these

earthquake sources are simulated and tsunami wave heights along the coast of Fethiye, Turkey are computed by Nami-Dance software (Yalciner et al., 2006).

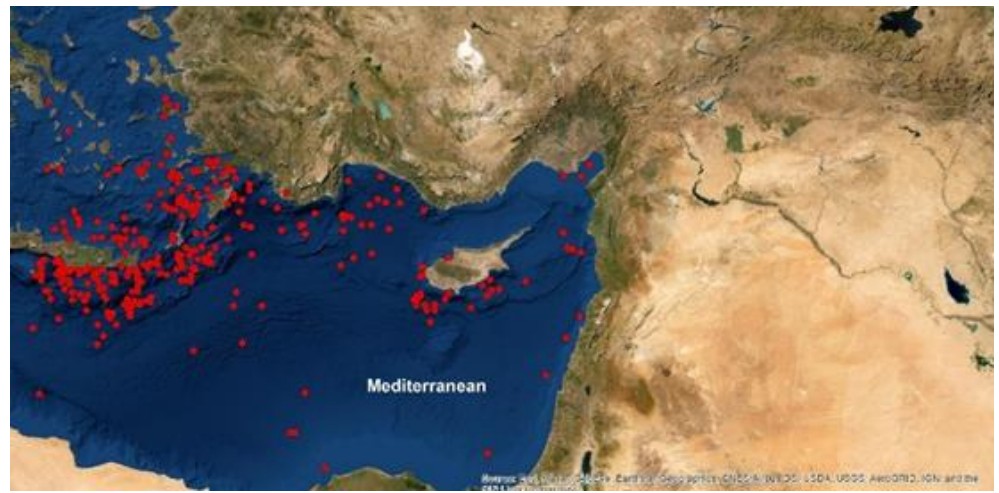

**Figure 7.** Historical earthquake locations that used as the epicenter of the hypothetical earthquakes (Source: Esri, Maxar, GeoEye, Earthstar Geographics, CNES/Airbus DS, USDA, USGS, AeroGRID, IGN, and the GIS User Community).

**2.2.     Tsunami Simulations**

100000 earthquake magnitudes are generated via Monte Carlo simulations and 1561 hypothetical earthquake sources having $M_w \geq 6.5$ are compiled to evaluate the flood and tsunami hazards simultaneously for the selected region based on the suggested framework by Yavuz et al., (2020). Bathymetry of the study area has a 407 m grid size is retrieved from the General Bathymetric Chart of the Oceans (GEBCO, n.d.). Nami-Dance software that runs the continuity and momentum equations as

shallow water equations is used to perform tsunami simulations to compute the tsunami wave height ($d_t$) at the coast of Fethiye, Turkey. The shallow water equations are expressed as follows (Velioglu et al., 2016):

$$\frac{\partial \eta}{\partial t} + \frac{\partial M}{\partial x} + \frac{\partial N}{\partial y} = 0 \tag{5}$$

$$\frac{\partial M}{\partial t} + \frac{\partial}{\partial x}\left(\frac{M^2}{D}\right) + \frac{\partial}{\partial y}\left(\frac{MN}{D}\right) + gD\frac{\partial \eta}{\partial x} + \frac{gn^2}{D^{7/3}}M\sqrt{M^2 + N^2} = 0 \tag{6}$$

$$\frac{\partial N}{\partial t} + \frac{\partial}{\partial x}\left(\frac{MN}{D}\right) + \frac{\partial}{\partial y}\left(\frac{N^2}{D}\right) + gD\frac{\partial \eta}{\partial y} + \frac{gn^2}{D^{7/3}}N\sqrt{M^2 + N^2} = 0 \tag{7}$$

$$M = u(h + \eta) = uD \tag{8}$$

$$N = v(h + \eta) = vD \tag{9}$$





where $\eta$ is the disturbance at the sea surface due to fault displacement, $t$ is time, $x$ and $y$ are the horizontal axes on the sea surface, $n$ is the Manning's roughness coefficient, $M$ and $N$ are the discharge fluxes, $D$ is the total sea depth, $g$ is the gravitational acceleration, $u$ and $v$ are the water particle velocities and $h$ is the undisturbed sea depth. Nami-dance software

has a capability to compute generation, propagation, and amplification of tsunami waves using the shallow water equations given above (Velioglu et al., 2016).

In this study, tsunami wave amplification cannot be calculated due to the coarse grid size of the bathymetry. Therefore, a commonly used empirical equation proposed by Green (Synolakis, 1991; Løvholt et al., 2012,2014; Yavuz et al., 2020) is used to calculate $d_t$ at 1 m water depth at the coast. To apply the equation, a gauge is digitized at 50 m water depth and Green's law

(Synolakis, 1991) is used to calculate $d_t$ at 1 m depth at the coast of the selected region.

$$d_t = \sqrt[4]{\frac{h_{50}}{h_1}} d_{50} \tag{10}$$

where $h_{50}$ and $h_1$ are the undisturbed water depths at 50 m and 1 m, respectively. $d_{50}$ is the tsunami wave height recorded at the digitized gauge point in the simulation. $d_t$ is used to determine the additional flooded lands resulting from the simultaneous occurrence of the flood and tsunami hazards at the selected regions. The hypothetical earthquakes having the annual exceedance probabilities from $10^{-4}$/year to $10^{-1}$/year are considered as the earthquakes that can generate a tsunami at the Fethiye

coastline. It is known that a tsunami has a wave period of a couple of minutes, while the river flood could be much longer. However, it should be noted here that tsunami hazard assumed to be occurred at the time of fully developed flood hazard condition in this study. By doing so, $d_t$ is considered only as a water level at the downstream boundary condition, it neither change with time nor the water level at the river mouths. Flood hazard analyses are conducted for the discharge having the recurrence period of 10 years ($Q_{10}$). $Q_{10}$ flood discharge is selected due to its higher chance of coincidence with a probable

tsunami event than other commonly used flood periods in the literature. Thus, the coincidence of the combination of these two hazards changes from $10^{-5}$/year to $10^{-2}$/year.

### 2.3. Hydrodynamic Modeling and Quantification of Flood Hazard

Flood hazard is also evaluated with and without the presence of earth-quake-triggered tsunamis. 1D and 2D hydraulic modeling of the streams within the Fethiye City center are conducted by implementing MIKE 11, MIKE 21 FM, and MIKE Flood widely

accepted and used software for simulating hydraulic engineering problems (DHI, 2016).

Firstly, 1D numerical modeling is conducted by MIKE 11 which solves Saint Venant's Equations (DHI, 2016). For this purpose, the physical conditions of each stream are determined by field trips. By using the Nivolman GPS device, the layout of cross-sections is determined at every 100 m for each stream. Moreover, the dimensions and locations of culverts or inline structures are determined at the field. Therefore, obtained data from the field are inserted into MIKE 11 to represent the real

physical conditions of the study area. Finally, a 1D numerical model via MIKE 11 is conducted and areas prone to flooding are determined by considering the bank elevations and water levels within each cross-section.




After having implemented the 1D numerical model, it is able to conclude that there is a possibility of flooding within the Fethiye City Center. Therefore, MIKE 21 FM model is implemented for the area of the city center. MIKE 21 is widely used software for modeling free-surface flows (DHI, 2016). The software solves shallow water equations which are incompressible

Reynolds averaged Navier-Stokes equations (DHI, 2016). Excess discharge within the stream bed (1D model) is released from the river banks and released to the surface thus numerical solution of surface water flows is implemented by MIKE 21. For this purpose, a digital elevation model (DEM) of the area with a resolution of 1 m is obtained from Fethiye Municipality. The DEM of the project area is illustrated in Figure 8.

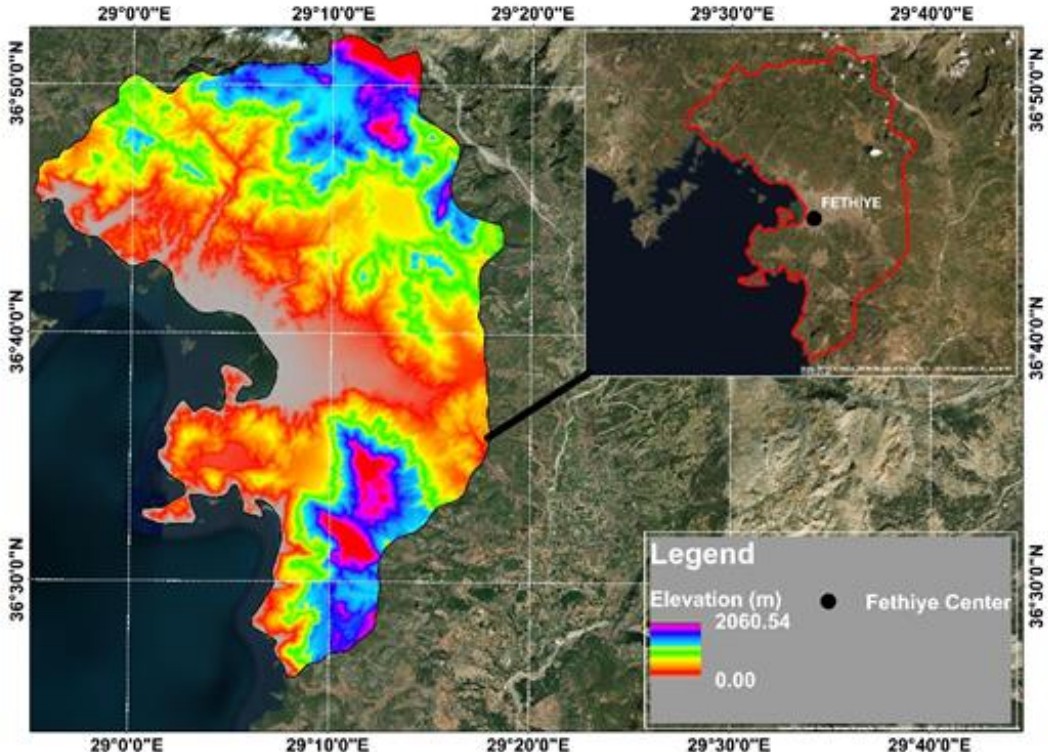

**Figure 8.** Demonstration of the DEM of study area (Source: Esri, Maxar, GeoEye, Earthstar Geographics, CNES/Airbus DS, USDA, USGS, AeroGRID, IGN, and the GIS User Community).

Both the 1D model and 2D model are coupled via MIKE Flood software, thus, excess discharge within the stream bed is released from the banks of the stream and the computational area is inundated. In order to solve the surface flow, the computational domain is meshed with non-uniform unstructured meshes. Moreover, the buildings/structures within the

computational area are digitized and implemented into MIKE 21 model to determine the area with fine meshes. The buildings within the computational area are excluded from the meshing procedure by considering the building elevations and possible inundation water levels. The result is provided by solving 1D and 2D numerical models simultaneously. The stream network of the selected region including Fethiye City Center is presented in Figure 9.





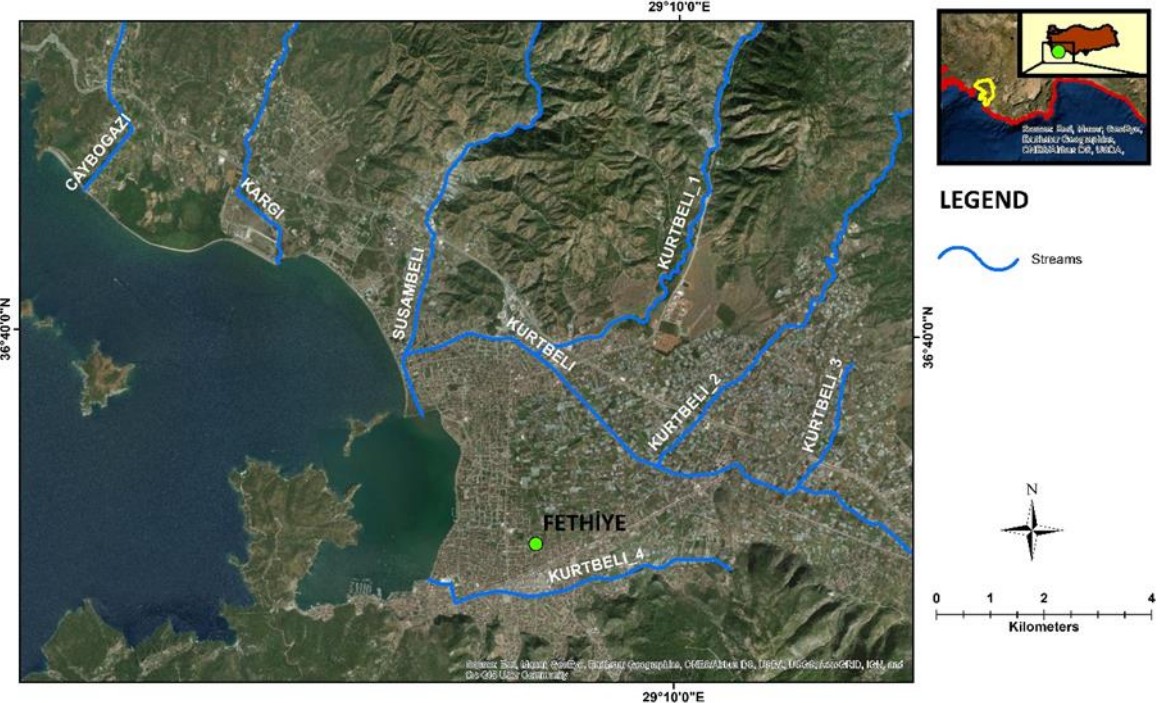

**Figure 9.** Stream network of the selected region (Source: Esri, Maxar, GeoEye, Earthstar Geographics, CNES/Airbus DS, USDA, USGS, AeroGRID, IGN, and the GIS User Community).

Throughout the simulations processes, input boundary conditions of each stream are determined as the discharge of 10 years of recurrence interval ($Q_{10}$). The calculated $Q_{10}$ discharges for each stream are tabulated in Table 1 and are provided from the "Hydrology Report" of "Flood Management Plan of Western Mediterranean Basin" which was prepared by the General Directorate of Water Management of Turkey under the guidance of "EU Flood Directive 2007/60" and "Water Framework Directive" (SYGM, n.d.).

**Table 1.** Peak Discharges of the Streams for Discharge of 10 Years Recurrence Interval in the Study Area (SYGM, n.d.).

| Fethiye City Center | | | |
|---|---|---|---|
| **Stream** | $Q_{10}$ **(m³/s)** | **Stream** | $Q_{10}$ **(m³/s)** |
| Caybogazi | 197.88 | Kurtbeli_2 | 11.28 |
| Kargi | 32.93 | Kurtbeli_3 | 4.56 |
| Kurtbeli | 24.44 | Kurtbeli_4 | 10.16 |
| Kurtbeli_1 | 19.11 | Susambeli | 58.2 |

The downstream boundary condition for a discharge of having 10 years return period of each stream is determined as water level. Moreover, calibration of the hydraulic model is not able to accomplish due to the lack of data. However, the most important parameter for calibrating the hydraulic model is manning's roughness coefficient. The surface roughness coefficients





are determined by considering CORINE 2018 Land Cover data (Papaioannou et al., 2018). The computational area was classified according to the land use classification of CORINE 2018 data as shown in Figure 10. Spatially varied roughness coefficients of the specific land cover were implemented according to the study conducted by Papaioannou et al., (2018).

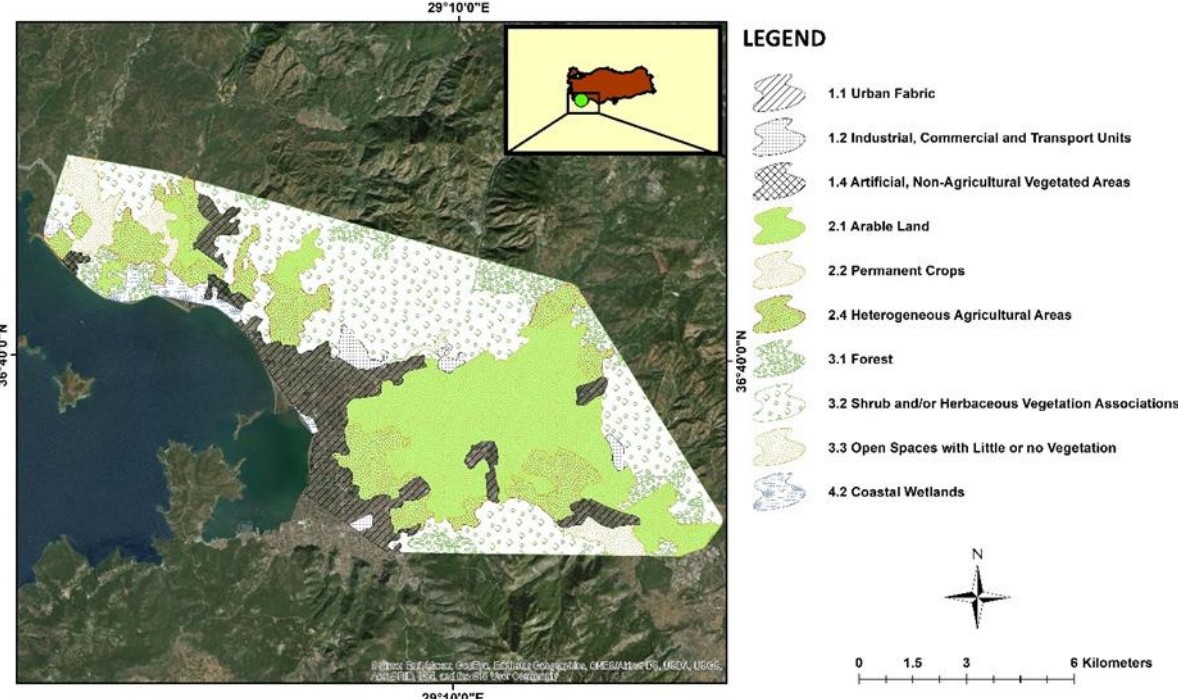

**Figure 10.** Land cover classification of computational domain according to CORINE 2018 Data (Source: Esri, Maxar, GeoEye, Earthstar Geographics, CNES/Airbus DS, USDA, USGS, AeroGRID, IGN, and the GIS User Community).

Average Manning's surface roughness coefficients study of each land cover of CORINE 2018 data was presented by Papaioannou et al., (2018). The land cover of the computational domain is constructed by examining the CORINE data and the roughness coefficients of each land cover are tabulated in Table 2.

**Table 2.** Peak Discharges of the Streams for Discharge of 10 Years Recurrence Interval in the Study Area (Papaioannou et al., 2018).

| Label 1 | Label 2 | Manning's n |
|---|---|---|
| 1 Artificial Surfaces | 1.1 Urban Fabric | 0.013 |
| | 1.2 Industrial, Commercial and Transport Units | 0.013 |
| | 1.3 Mine, Dump and Construction Sites | 0.013 |
| | 1.4 Artificial, non-Agricultural Vegetated Areas | 0.025 |
| 2 Agricultural Areas | 2.1 Arable Land | 0.03 |
| | 2.2 Permanent Crops | 0.08 |





| Label 1 | Label 2 | Manning's n |
|---|---|---|
| | 2.3 Pastures | 0.035 |
| | 2.4 Heteregenous Agricultural Areas | 0.045 |
| 3 Forest and Semi Natural Areas | 3.1 Forests | 0.1 |
| | 3.2 Scrub and/or herbaceous Vegetation Associations | 0.04 |
| | 3.3 Open Spaces with little or no Vegetation | 0..025 |
| 4 Wetlands | 4.1 Inland Wetlands | 0.04 |
| | 4.2 Coastal Wetlands | 0.04 |
| 5 Water Bodies | 5.1 Inland Waters | 0.05 |
| | 5.2 Coastal Waters | 0.07 |

After having carried out the hydraulic analysis, the result of the model is also used for flood hazard quantification. Flood hazard quantification is often conducted by considering water depth and velocity. Although there are various methods for quantifying flood hazards, direct multiplication of depth and velocity is suggested by Smith et al., (2014). The thresholds values for each hazard class and vulnerability classification are tabulated in Table 3 below (Smith et al., 2014).

**Table 3.** Hazard Classes and Vulnerability Thresholds (Smith et al., 2014).

| Hazard Vulnerability Classification | Description | Classification Limit ($m^2$/s) |
|---|---|---|
| H1 | Generally safe for vehicles, people and building | $D*V \leq 0.3$ |
| H2 | Unsafe for small vehicles | $D*V \leq 0.6$ |
| H3 | Unsafe for vehicles, children and the elderly | $D*V \leq 0.6$ |
| H4 | Unsafe for vehicles and people | $D*V \leq 1$ |
| H5 | Unsafe for vehicles and people. All buildings vulnerable to structural damage. | $D*V \leq 4$ |
| H6 | Unsafe for vehicles and people. All building vulnerable to failure | $D*V \leq 4$ |

Water depth within the inundated area and flood propagation velocity are both considered with and without the presence of an earthquake-triggered tsunami. Therefore, spatially varied hazard maps are constructed accordingly.

## 3. Results and Discussions

In this study, potential combined hazard assessment because of flood hazard ($Q_{10}$) with and without the presence of earthquake-triggered tsunamis are analyzed for Fethiye city center. Inundated areas due to flood only, earthquake-triggered tsunami only, and combined hazard (i.e. flood+earthquake-triggered tsunami) are determined by numerical computations and corresponding inundation levels are revealed for each hazard circumstance.




By considering only flood hazard having 10 years return period, maximum water levels are observed within the riverbed. The
inundated area due to flood is limited along the streamlines for inland sections. There are also small inundated sections that
can be observed due to flood at some parts of the coast of the study area. A large portion of the coastal region is not affected
by the flood waves and the inundated area is limited in the coastal parts. A sample inundation map of the study area is given
in Figure 11 for the flood of $Q_{10}$ obtained from the numerical computations.

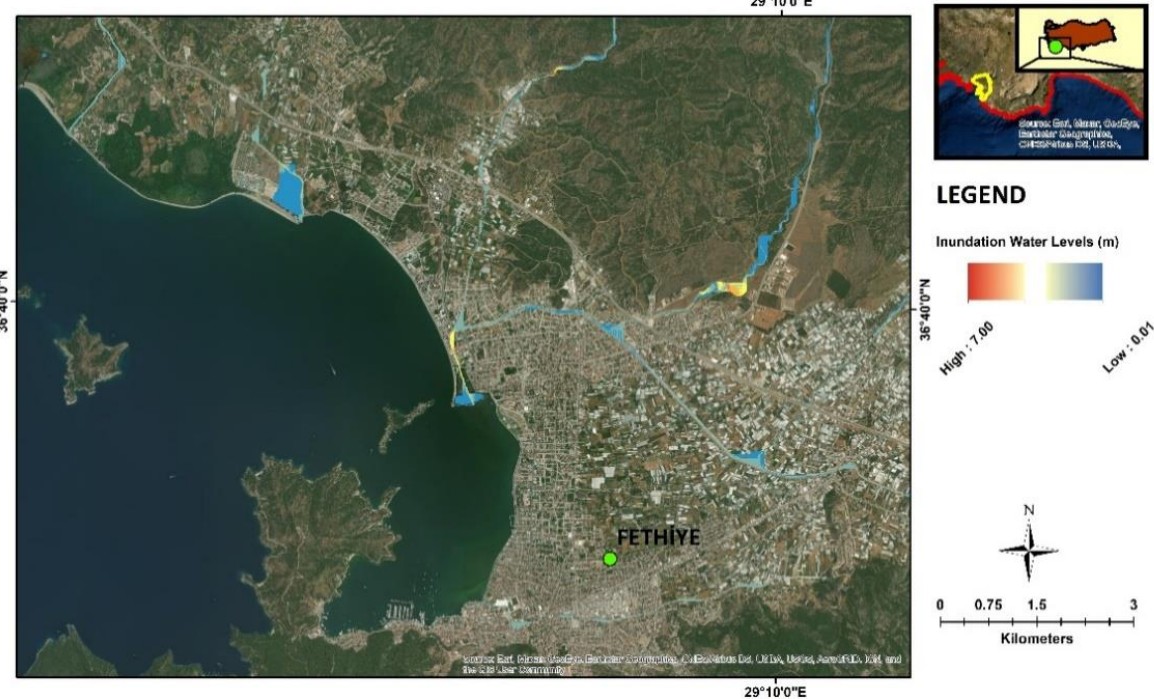

**Figure 11.** Inundation due to flood hazard (Source: Esri, Maxar, GeoEye, Earthstar Geographics, CNES/Airbus DS, USDA,
USGS, AeroGRID, IGN, and the GIS User Community).

Although exceeding 6 m water inundation level is observed on some parts of Kurtbeli_1 stream, the effect of $Q_{10}$ flood is
limited at the coastline. Depending on the computation results, the other streams also have small inundations around the river
beds.

For earthquake-triggered tsunami hazard condition, significant portion of the coastline estimated to be inundated with 3.5 m
tsunami wave heights (see Figure 12). Comparing with the flood hazard level, earthquake-triggered tsunamis might have
considerable inundation levels at the coastline. Reaching up to 1 km of land from the coastline is estimated to be inundated
due to tsunami waves depending on the hypothetical earthquake-triggered tsunami analysis.


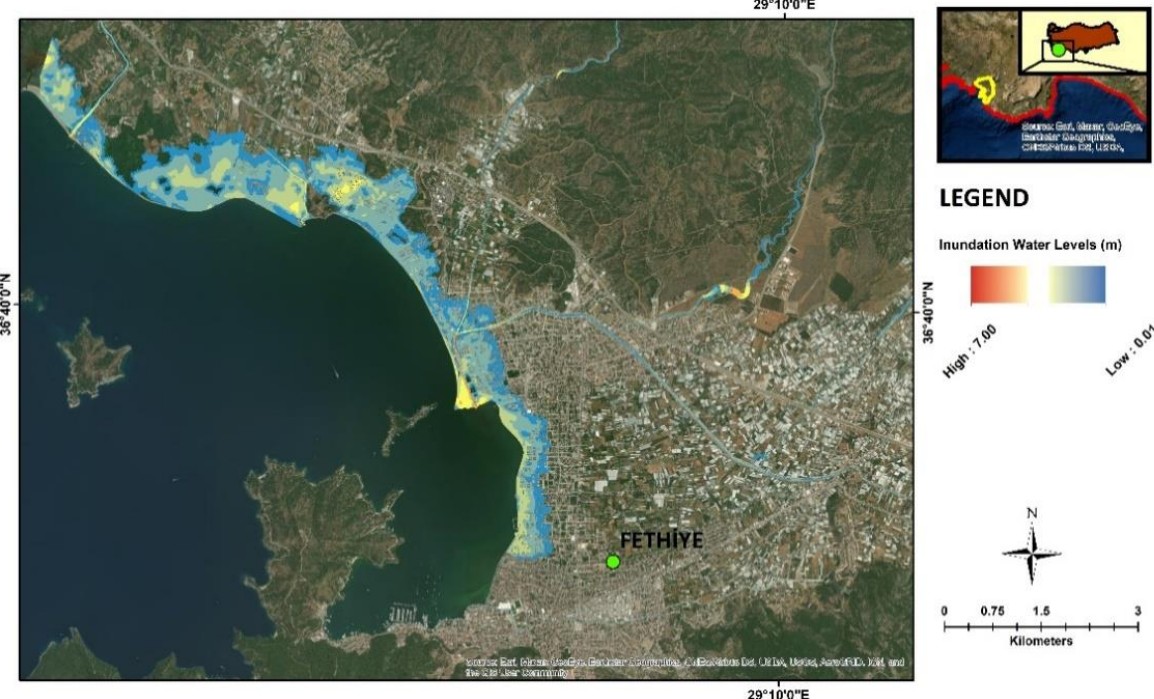

**Figure 12.** Inundation levels resulted from an earthquake-triggered tsunami hazard (Source: Esri, Maxar, GeoEye, Earthstar Geographics, CNES/Airbus DS, USDA, USGS, AeroGRID, IGN, and the GIS User Community).

On the other hand, the coastline of the study area is severely inundated due to flood which was take place slightly before tsunami peak waves hit the coastal parts of the city. Although the maximum tsunami wave height obtained from the simulations is around 3.50 m, the inundation level for the combined hazard existence reaches up to 7.00 m for some parts of the low-lying sections of the study area (see Figure 13).


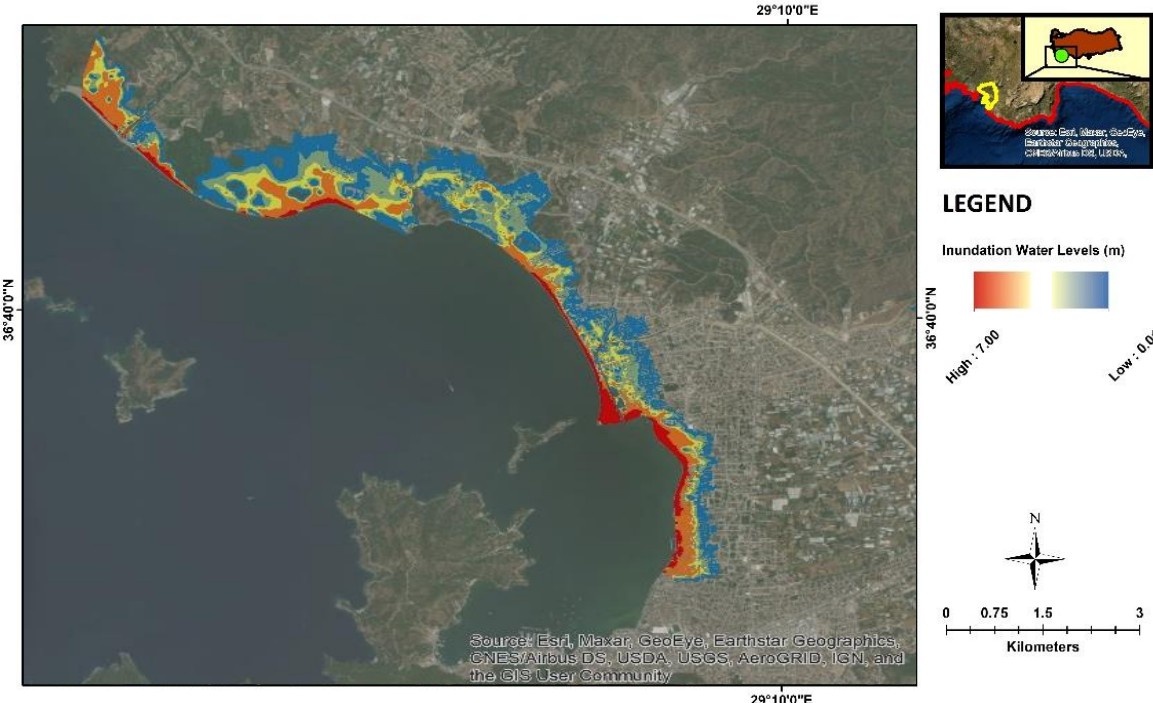

**Figure 13.** Inundation levels obtained from the simultaneous occurrence of fully developed flood and earthquake-triggered tsunami hazards (Source: Esri, Maxar, GeoEye, Earthstar Geographics, CNES/Airbus DS, USDA, USGS, AeroGRID, IGN, and the GIS User Community).

Quantification of flood hazard is also carried out by considering the threshold values and classes given in Table 3. Results of hazard quantification is presented in Figure 14.




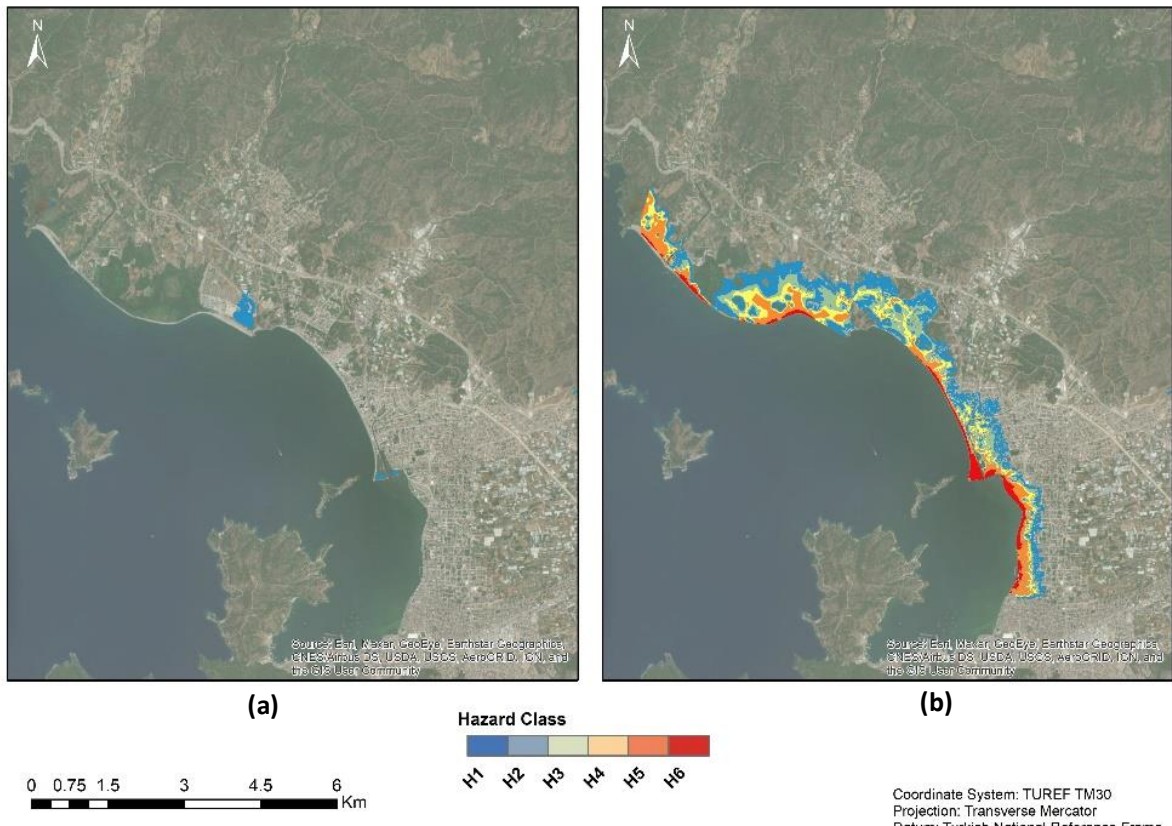

**Figure 14.** Spatially varied hazard mapping for (a) flood and (b) flood+earthquake-triggered tsunami (Source: Esri, Maxar, GeoEye, Earthstar Geographics, CNES/Airbus DS, USDA, USGS, AeroGRID, IGN, and the GIS User Community).

According to hazard vulnerability classification proposed by Smith et al., (2014), flood of 10 years' recurrence interval drop into H1 class hazard, which can be considered insignificant, in some of the coastal parts of the city center. On the other hand, the fully developed flood of 10 years of recurrence interval just after peak tsunami waves reach the coast resulted in varying hazard classes of H1 to H6. The major portion of the hazard is caused by the tsunami.

**4. Conclusions**

Flood and potential earthquake-triggered tsunami hazards are simultaneously analyzed to evaluate the amount of inundation levels at the coastline of Fethiye Bay and Fethiye city center. Results demonstrate that majority of the increase in inundation levels is due to tsunami hazard. However, it should be emphasized that inundation levels are almost doubled in the presence of flood hazard at the same time. In the analyses, it is assumed that fully developed flood take place just after the peak tsunami waves hit the coastal region. Therefore, sea levels are determined accordingly for the hydraulic models.



Flood of 10 years of recurrence period was taken into consideration in the study and potential hazards are calculated. Although it is more sophisticated to reduce the effects of tsunamis, prevention of floods as well as their consequences is a more common procedure. Thus, combined risk analyses of multiple hazards should be taken into consideration in order to reduce risks due to natural disasters.

In conclusion, the coincidence of flood and tsunami events might have a very low chance. But the combination of these two
hazards is definitely increased the inundation levels and corresponding disaster levels in the selected region. Some other factors such as seasonal changes in economic and social aspects, the expansion of the residential sites, proximity to the fault zones, and climate change effects should be taken into consideration in combined risk analysis for future years.

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
