# Peer review of "Combined Hazard Analysis of Flood and Tsunamis on The Western Mediterranean Coast of Turkey"

_Natural Hazards and Earth System Sciences, 2022_

## Author Response (AR1)

**RESPONSE TO REVIEWERS COMMENTS:**

*Authors' General comment: We would like to thank to the reviewers for constructive comments. We have responded to the reviewers' comments and we have made required revisions on the manuscript.  We believe the revised version of the manuscript, which addresses the reviewers' comments, is now more consistent with the current literature and clarifies the important points raised by the reviewers. With the changes, the manuscript is re-submitted in **clean** format to the Journal. Please also find below my response to reviewers' comments.*

**REVIEWER 1:**

**GENERAL COMMENTS:** In the manuscript (nhess-2022-121), the potential effects of tsunami and flood hazards to the residential region of the west part of the Mediterranean Sea are examined numerically. Multi-hazard analyses of flood and tsunami events are rare in the literature. Hence, the research is notable in terms of taking proper precautions against marine-caused natural hazards and ensure population safety. The methodology proposed in the manuscript has a potential to motivate researchers to conduct similar studies. The manuscript is well organized and clear. The manuscript is of interest to NHESS Journal's readers. I believe that the manuscript will be much better if the points raised below are revised.

*AUTHORS RESPONSE: Authors appreciate the constructive comments of this reviewer. The objective of the manuscript is clearly enlightened by the reviewer.*

**REVIEW COMMENTS:**

1.1. Line 1: Combination of flood and tsunami hazards may not possible in general. Considering the flow of the manuscript content, a multi-hazard analysis of flood and tsunami definition is more proper than the definition of a combination of these two hazards. Please change the "combined hazard analysis" with "multi-hazard analysis" in the text?

*AUTHORS RESPONSE: We appreciate the reviewer's comments. All the definitions mentioning the bilateral analysis of tsunami and flood hazards are replaced with "multi-hazard" instead of "combined hazard".*

1.2. Line 48: Flood hazard can be defined as epistemic uncertainty as mentioned in the abstract. Please fix the definition of stochastic analysis of flood as epistemic uncertainty instead of aleatory variability.

*AUTHORS RESPONSE: We appreciate the reviewer's comments. In L46-L48, "The exceedance of flood hazard is strongly likely depending on geological and meteorological circumstances, the hazard is included in the stochastic analyses conducted in this study as aleatory variability" **sentence is modified as** "The exceedance of flood hazard is strongly likely depending on geological and meteorological circumstances, the hazard is included in the stochastic analyses conducted in this study as epistemic uncertainty".*

1.3. Line 49: Tsunami hazard can be defined as aleatory variability by focusing on the exceedance probabilities in the tsunami hazard curves. In the abstract, it is mentioned that statistical analysis of tsunamis is conducted as aleatory variability. Please change the sentence accordingly.

**AUTHORS RESPONSE:** *We appreciate the reviewer's comments. In L48-L49, "Since the occurrence of the tsunami is generally rare compared with flood hazards, tsunami events are inspected by considering epistemic uncertainty in this study"* **sentence is modified as** *"Since the occurrence of the tsunami is generally rare compared with flood hazards, tsunami events are inspected by considering aleatory variability in this study".*

1.4. Line 91: Statistical approach should also be aleatory variability for tsunamis.

**AUTHORS RESPONSE:** *We appreciate the reviewer's comments. In L91, we made the required change.*

1.5. Line 107: hypocenter should be replaced with hypocenter distance.

**AUTHORS RESPONSE:** *We appreciate the reviewer's comments. In L107, we added the required word.*

1.6. Line 107: One of the reasons of dip angle assignment should be the width of the fault (W). Please explain the reason of assigning dip angles as mentioned in the text clearly to the hypothetical earthquake sources"

**AUTHORS RESPONSE:** *We appreciate the reviewer's comments. We appreciate the reviewer's comments. In L107, "Depending on the hypocenter distance of the hypothetical earthquake, dip angles are assigned as $30^0$,$60^0$, and $90^0$"* **sentence is modified as** *"In this study, the asperity position of the hypocenter is assumed to be at the center of the fault and hypocenter distances are directly obtained from the historical earthquake dataset. In some circumstances, hypocenter distances are smaller than the calculated W values. This phenomenon causes some problematic solutions. To prevent this kind of miscalculations, dip angles are randomly assigned as $30^0$,$60^0$, and $90^0$ to the grouped hypocenter distances considering the W values as well.".*

1.7. Figure 7: The quality of Figure 7 should be improved.

**AUTHORS RESPONSE:** *We appreciate the reviewer's comments. Resolution of the figure is enhanced as mentioned.*

1.8. Figure 8: Please improve the resolution of the Figure.

**AUTHOR'S RESPONSE:** *We appreciate the reviewer's comments. Resolution of the figure is enhanced as mentioned.*

**REVIEWER 2:**

***GENERAL COMMENTS:*** The paper presents well the models and methods used, and the results it presents are clear and well presented. But to be published, I would expect a deeper investigation of the multi-hazards and their interaction. At least 1 or 2 experiments for example:
It would be interesting to rerun the model with a range of return levels (you mention 50-year and 100-year floods in your introduction)
In the model, you assumed that the river flood occurs just after the peak tsunami. I would like to see further experiments where you explore the sensitivity of the timing of the flood (with and without the sea levels change introduced to the hydraulic model by tsunami)
There is some good material here - and I appreciate the work that has gone into preparing the models and data. Performing some more experiments to explore the sensitivities to these multi-hazards would greatly inform the quality of the paper and scientific interest namely:
- experiments with river flood of different return periods (Q50, Q100)
- experiments with flood arriving with/without tsunami influenced sea-level
I recommend a major revision before this paper is acceptable for publication. Wish you the best in developing your interesting work.

***AUTHORS RESPONSE:*** *Authors appreciate the constructive comments of this reviewer. The objective of the manuscript is clearly enlightened by the reviewer.*

**REVIEW COMMENTS:**

1.1. It would be interesting to rerun the model with a range of return levels as you mentioned 50-year and 100-year floods in your introduction?

***AUTHORS RESPONSE:*** *We appreciate the reviewer's comments. Simulations with Q50, Q100 as well as their effect with tsunami were implemented and results were discussed as per comments. In lines 76-80, "Flood hazard having the recurrence period of 10 years ($Q_{10}$), on the other hand, is modeled by MIKE 11, MIKE 21 FM and MIKE Flood considering with and without tsunami wave existence at the coasts (DHI, 2016). As a more frequent flood period, $Q_{10}$ is evaluated in this study instead of the flood events having the return period of 50 years or 100 years"* **sentence is modified as** *"Flood hazards having the recurrence period of 10, 50, and 100 years ($Q_{10}, Q_{50}, and\ Q_{100}$), on the other hand, is modeled by MIKE 11, MIKE 21 FM and MIKE Flood considering with and without tsunami wave existence at the coasts (DHI, 2016). As a more frequent flood period, $Q_{10}$ is evaluated in detail and hazard maps are generated for the flood events having the return period of 50 years ($Q_{50}$ ) and 100 years ($Q_{100}$,) respectively. Additionally, tsunami-drifted flood hazard levels are also provided for all three flood events to satisfy the multi hazard assessment procedure presented in this study". All related results and discussions about the newly added events are presented in the manuscript.*

1.2. In the model, you assumed that the river flood occurs just after the peak tsunami. I would like to see further experiments where you explore the sensitivity of the timing of the flood (with and without the sea levels change introduced to the hydraulic model by tsunami. Performing some more experiments to explore the sensitivities to these multi-hazards would greatly inform the quality of the paper and scientific interest namely experiments with flood arriving with/without tsunami influenced sea-level

*AUTHORS RESPONSE: We appreciate the reviewer's comments.* Flood hazards having the recurrence period of 10, 50, and 100 years ($Q_{10}, Q_{50}, and\ Q_{100}$), are analyzed with and without earthquake-triggered tsunami as deeply investigated in the manuscript. Figure 10 and Figure 13 demonstrate the inundated lands due to flood only and multi-hazard conditions. *Figure 11 and Figure 12 is also illustrate the effect of tsunami only and flood event ($Q_{10}$) + tsunami inundation levels. By conducting new experiments and analyses, the authors believe that the sensitivity of the multi-hazard assessment is enhanced in the manuscript.*

1.3. When you say 'flood' this is fluvial only? Not precipitation or coastal (wave/surge). You mention you are not considering hazard resulting from seismicity - but also explain if/why precipitation and coastal wave and surge floods are out of scope. e.g. could you give a rough figure on how much % of freshwater flood is driven by river v.s. rain in this region to easily justify omission?

Turkey is at high risk from surge / coastal flood

https://www.thinkhazard.org/en/report/249-turkey/CF

*AUTHORS RESPONSE: We appreciate the reviewer's comments. Only fluvial hazard is considered one of the multi-hazard assessments conducted in this study. Any kind of fluvial hazard resulting from precipitation, groundwater contribution, etc. can be evaluated as the fluvial hazard event conducted in this study. Thus, flood hazards resulting from storm surges or precipitation etc. do not considered as the flood hazard in this study. The discharges carrying by the rivers in the study area are given in Table 1 for all three flood events having different return periods. To clarify the flood events considered in this study, the sentence in line 155 "Flood hazard is also evaluated with and without the presence of earthquake-triggered tsunamis"* **is modified as** *"Fluvial hazards resulting from the water level rise in the river and overflow onto the neighboring lands are also evaluated considering three different return periods with and without the presence of earthquake-triggered tsunamis".*

1.4. Do you need Figure 1? Just taken from someone else's paper) Maybe just add a line in the text summarizing Munich Re (2020)

*AUTHORS RESPONSE: We appreciate the reviewer's comments. Depending on the reviewer's comment on Figure 1, the figure removed from the paper and the following sentences are added to clarify the information retrieved from Munich RE.*
*"Munich RE (2020) has set a natural catastrophe loss database on natural disasters since 1980s for analyzing and assessing losses resulted by natural disasters. The database revealed that number of floods and their destructive economic results have an upward trend at global scale."*

1.5. Figure 2: consider showing epicentres of earthquakes in your scene-setting map. Highlight areas of flooding - this is a missed opportunity to share more info in the figure. Actually, this information is present in subsequent figures - therefore suggest you just cut figure 2 as well.

*AUTHORS RESPONSE: We appreciate the reviewer's comments. The authors are aimed to give a magnified location of the study area and its surroundings in Figure 2 (newly Figure 1). Figure 2 contains some other information like residential locations around the study area and their approximate distances. The subsequent figures do not contain similar information but focused on specific conditions and results of the*

*analyses. In order not to create any confusion, the authors kindly aim to hold Figure 2 (newly Figure 1) in the paper as it is.*

1.6. Gutenberg-Richter relationship is a key method. Introduced on p3 line 65, but not further defined. Please include the equation and a reference at this point - I presume it's this?? "Gutenberg, B., Richter, C. F., 1956. Magnitude and Energy of Earthquakes. Annali di Geofisica, 9: 1–15.

**AUTHORS RESPONSE:** *We appreciate the reviewer's comments. In lines 66-73,* "*The Gutenberg-Richter relationship is a mathematical expression of the relationship between a number of earthquakes and the Richter magnitudes (M_w) of these earthquakes that occurred in a specific region (Gutenberg & Richter, 1956). They proposed a widely accepted and commonly used empirical equation that explains the relationship between the occurrence probability of an earthquake depending on two seismic constants (i.e. a and b values) which define the frequency-magnitude distribution and the Richter magnitudes experienced in a particular region. The equation is defined as follows:*

$$\log N = -\mathrm{b}M_w + \mathrm{a} \tag{1}$$

*where N is the number of earthquakes experienced in the selected region, a and b are the constant that defined specifically for the selected region". **sentences are added** to clarify and define the Gutenberg-Richter relationship in detail using the following reference.*

Gutenberg, B., & Richter, C. F. (1954). Seismicity of the Earth 2nd ed., 310 pp. Princeton University Press, Princeton, New Jersey.

**TECHNICAL CORRECTIONS:**

1.7. Line 46-48 long sentence - not clear what you are saying. Is exceedance a specific statistical term? "The exceedance of flood hazard is strongly likely depending on geological and meteorological circumstances, the hazard is included in the stochastic analyses conducted in this study as aleatory variability"

**AUTHOR'S RESPONSE:** *We appreciate the reviewer's comments. Exceedance is a specific term commonly used in statistical analyses. The probability of exceedance for every single event in statistical analyses is inspected to evaluate the adverse effects of the incidences. The confusion on the sentence is also mentioned by Reviewer 1. Therefore, In L46-L48, "The exceedance of flood hazard is strongly likely depending on geological and meteorological circumstances, the hazard is included in the stochastic analyses conducted in this study as aleatory variability" **sentence is modified as** "The exceedance of flood hazard is strongly likely depending on geological and meteorological circumstances, the hazard is included in the stochastic analyses conducted in this study as epistemic uncertainty".*

1.8. Figure 4: please label axes so that it is clear which magnitude is generated by which method (Gutenberg-Richter vs. normal) - currently they are both labelled as "M$_w$"

**AUTHOR'S RESPONSE:** *We appreciate the reviewer's comments. The labels of Figure 4 (newly Figure 3) is revised as $M_{w(G-R\ relationship)}$ for y axis and $M_{w(normally\ distributed)}$ for x axis, respectively.*

1.9. Figure 5: what are Sample 1,2,3 ? This is not explained.

**AUTHOR'S RESPONSE:** *We appreciate the reviewer's comments. Starting from line 99, "*Tsunami hazards curves are derived to determine the reliability of Monte Carlo simulations by considering the aleatory variability of each hypothetical earthquake magnitude by checking the consistency of the curves*" **sentence is modified as** "*Three different tsunami hazards curve samples that derived from 100000 Monte Carlo simulations are used to determine the reliability of Monte Carlo simulations by considering the aleatory variability of each hypothetical earthquake magnitude by checking the consistency of the curves. The curve samples are shown as Sample_1, Sample_2 and Sample_3 in Figure 4.*" *to clarify the definition of samples given in Figure 5 (newly Figure 4).*

1.10. The legend of figure 6 is wrong (repeat of figure 5 legend).

**AUTHOR'S RESPONSE:** *We appreciate the reviewer's comments. The authors are extremely sorry due to the mistake. The legend of Figure 6 (newly Fıgure 5) is revised as "Generation steps of the hypothetical earthquake sources".*

1.11. line 178 : "The downstream boundary condition for a discharge of having 10 years return period of each stream is determined as water level. " Should this be 'determined as mean sea-level"?

**AUTHOR'S RESPONSE:** *We appreciate the reviewer's comments. The water level term is changed as mean sea-level in the manuscript. Since the software used to simulate the multi-hazard condition considers the downstream boundary level as water level, authors would like to express the software's term as given in the software.*